# Diagnostic Value of Four-Dimensional Dynamic Computed Tomography for Primary Hyperparathyroidism in Patients with Low Baseline Parathyroid Hormone Levels

**DOI:** 10.3390/diagnostics13162621

**Published:** 2023-08-08

**Authors:** Zaid Al-Difaie, Max H. M. C. Scheepers, Sanne M. E. Engelen, Bastiaan Havekes, Nicole D. Bouvy, Alida A. Postma

**Affiliations:** 1GROW School for Oncology and Developmental Biology, Maastricht University, 6229 ER Maastricht, The Netherlands; z.al-difaie@maastrichtuniversity.nl (Z.A.-D.); m.scheepers@maastrichtuniversity.nl (M.H.M.C.S.); 2Department of Surgery, Maastricht University Medical Center, 6202 AZ Maastricht, The Netherlands; 3Department of Internal Medicine, Division of Endocrinology and Metabolism, Maastricht University Medical Center, 6202 AZ Maastricht, The Netherlands; 4Department of Radiology and Nuclear Medicine, School for Mental Health and Sciences (MHENS), Maastricht University Medical Center, 6202 AZ Maastricht, The Netherlands

**Keywords:** primary hyperparathyroidism, four-dimensional computed tomography, parathyroid hormone, diagnostics

## Abstract

Low baseline levels of parathyroid hormone (PTH) are associated with a higher rate of multiglandular disease, lower localization rates of preoperative imaging modalities, and a higher rate of unsuccessful minimally invasive parathyroidectomies. The objective of this study is to assess the diagnostic value of four-dimensional dynamic computed tomography (4D-CT) in localizing primary hyperparathyroidism (pHPT) in patients with low baseline PTH levels, compared to patients with high baseline PTH levels. Patients with pHPT who received a 4D-CT scan as part of their standard diagnostic evaluation were divided into two groups based on the following criteria: (1) preoperative PTH levels less than 100 pg/mL and (2) patients with preoperative PTH levels greater than 100 pg/mL. All patients underwent parathyroidectomy based on 4D-CT findings, with intraoperative parathyroid hormone monitoring. The lesion-based sensitivity of 4D-CT was 88% in patients with low baseline PTH levels and 94.7% in patients with high baseline PTH levels (*p* = 0.33). However, the success rate of image-guided resection based on 4D-CT findings was 71.4% in the low baseline PTH group compared to 90.6% in the high baseline PTH group (*p* = 0.06). Our study demonstrated that 4D-CT has a high lesion-based sensitivity in patients with pHPT and low baseline PTH levels but led to a relatively low rate of successful image-guided resection in patients with low baseline PTH levels. Therefore, it is important to exercise increased caution during 4D-CT-guided surgical exploration of patients with low baseline PTH levels to ensure successful surgical resection of all parathyroid lesions.

## 1. Introduction

The initial diagnosis of patients with primary hyperparathyroidism (pHPT) typically starts with the identification of elevated serum calcium and parathyroid hormone (PTH) levels [1]. However, patients with pHPT may present with different levels of serum PTH and calcium, which may lead to diverse clinical presentations [1,2,3]. Recognizing the diverse clinical presentations of pHPT is crucial for tailoring appropriate diagnostic and treatment strategies for individual patients.

Due to increasing neck imaging studies, and calcium measurements for osteoporosis screening, a growing number of patients with pHPT are diagnosed with low baseline levels of PTH [4]. Several studies have shown that low baseline levels of PTH (<100 pg/mL/10.6 pmol/L) are associated with distinct characteristics including a higher rate of multiglandular disease (MGD) [3,5,6], a higher rate of unsuccessful minimally invasive parathyroidectomies, and lower localization rates of preoperative imaging modalities [2,7,8]. Thus, accurate preoperative imaging techniques for the localization of parathyroid lesions are critical for surgical planning in this patient cohort with low baseline levels of PTH. Traditionally, preoperative localization has been performed with ultrasonography (US) and sestamibi scintigraphy (MIBI), which have a high diagnostic accuracy in most pHPT cases. However, in patients with multiple adenomas, recurrent disease, and ectopic adenomas, these imaging modalities can be less accurate [9].

Computed tomography (CT) has been studied extensively for the localization of pHPT due to its high spatial resolution, short acquisition time, and relatively low cost. This has led clinicians to investigate several different cross-sectional CT imaging techniques for the localization of pHPT, such as four-dimensional dynamic computed tomography (4D-CT) [10,11]. This imaging method uses multiphase scanning to visualize the unique perfusion characteristics of parathyroid lesions to detect parathyroid adenomas [12]. Four-dimensional dynamic CT has been established as an accurate imaging modality for the localization of pHPT [13,14,15,16,17,18,19,20,21]. Several studies have reported superior diagnostic accuracy of 4D-CT, including in patients with MGD and patients with recurrent pHPT, compared to traditional first-line imaging modalities such as ultrasound and sestamibi scintigraphy (MIBI) [13]. Furthermore, 4D-CT is the preferred imaging modality for localizing ectopic adenomas compared to US and MIBI [22,23]. Therefore, 4D-CT might be a promising pre-operative imaging modality for patients with low baseline PTH levels. 

There is a paucity of data regarding the diagnostic value of 4D-CT in patients with pHPT and low baseline PTH. Therefore, this study aims to evaluate the diagnostic value of 4D-CT in localizing pHPT in patients with low baseline PTH levels, compared to patients with high baseline PTH levels. 

## 2. Materials and Methods

### 2.1. Patients

This study was a retrospective analysis that included patients with pHPT who received a 4D-CT scan as part of their standard diagnostic evaluation followed by parathyroidectomy at the Maastricht University Medical Center in the Netherlands between June 2016 and August 2022. The Maastricht University Medical Center (The Netherlands) is a specialized center for the management of pHPT, serving as a tertiary referral center. The study protocol was approved by the Medical Ethical Committee (METC-17-4-077.2). All patients included in the study were confirmed to have pHPT through biochemical testing. Patients with multiple endocrine neoplasia (MEN 1 and MEN2) syndrome and secondary or tertiary hyperparathyroidism were excluded. Data regarding clinical and demographic patient characteristics were obtained from electronic medical records. Patients were divided into two study groups based on the following criteria: (1) preoperative PTH levels less than 100 pg/mL and (2) patients with preoperative PTH levels greater than 100 pg/mL.

### 2.2. Imaging Technique

Four-dimensional dynamic CT was performed on a third-generation dual source CT scanner (Somatom Definition Force, Siemens, Erlangen, Germany). The studied protocol included four phases: (1) a DECT or SECT non-contrast scan, (2) a DECT post-contrast scan at 30 s (‘arterial phase’), (3) a DECT 50 s post-contrast (‘venous phase’), and (4) a SECT post-contrast scan at 90 s (‘delayed phase’). Post-contrast scans were conducted after injection of 70 mL Ultravist 300 (Bayer Schering) with a flow rate of 3 mL/s, followed by 30 mL saline flush at the same flow rate. SECT scans were scanned in a care-kV energy mode. DECT scan parameters were as follows: tube voltages 80/150 kVp, Quality Reference mAs 35/23 mAs, CTDIVol 5.6 mGy. Rotation time 500 ms, pitch 0.7 and collimation 0.6, slice thickness 1 mm, FOV 270 mm, 512 × 512 matrices. All 4D-CT studies were primarily read or reviewed by a board-certified senior neuroradiologist (A.P). An example of a parathyroid adenoma detected with 4D-CT using 4 phases is provided in Figure 1.

### 2.3. Surgery

All patients underwent elective parathyroidectomy at the Maastricht University Medical Center+ (The Netherlands). All surgical approaches were based on the findings of 4D-CT, and exploration started at the side of the candidate lesion on imaging. Intraoperative parathyroid hormone (IOPTH) monitoring was used in all patients. A pre-excision blood sample was sent for baseline IOPTH level. After excision of the suspected adenoma(s), 5 and 10 min IOPTH post-excision levels were drawn. A drop of 50% and resulting normal IOPTH level 10 minutes post-excision were considered evidence of successful excision of parathyroid adenomas. The criteria for surgical cure were defined as a greater than 50% decrease in intraoperative PTH level from the highest baseline or pre-excision PTH level 10 min after parathyroidectomy, postoperative normalization of PTH/calcium level at first follow-up, and pathological confirmation of abnormal parathyroid tissue at pathologic examination. Serum PTH and calcium were drawn preoperatively and postoperatively. In our institution, the normal range of serum calcium was 8.42 mg/dL (2.10 mmol/L) to 10.22 mg/dL (2.55 mmol/L), and PTH was 12.26 pg/mL (1.3 pmol/L) and 64.12 pg/mL (6.8 pmol/L). Intact PTH was measured with a chemiluminescent immunometric assay (Immulite XPi instrument, Siemens Healthcare Diagnostics, New Orleans, LA, USA), and IOPTH was measured with the Immulite 1000 assay (Siemens Healthcare Diagnostics, New Orleans, LA, USA). All surgical specimens were evaluated by a trained pathologist, and weight, measurements, and histopathological features were recorded. 

### 2.4. Statistical Analysis

Lesion-based sensitivity was based on three potential lesion locations: left neck, right neck, and mediastinum. The right and left neck locations included ectopic glands. Lesions were only included in the lesion-based analysis if there was pathological confirmation of parathyroid adenomas or hyperplasia. Therefore, patients in which no lesion was found during surgery, were excluded from this analysis. Sensitivities for adenoma localization in 4D-CT were generated based on the original radiology reports and surgical records. 

On a patient-based level, successful image-guided surgery was defined as the excision of an abnormal parathyroid gland or glands on the same side (left, right, and mediastinum) as the 4D-CT imaging result accompanied by a reduction in PTH level of 50% from the highest baseline or pre-excision PTH level, pathological verification of parathyroid adenoma or hyperplasia, and postoperative normalization of PTH/calcium level at first follow-up. Conversely, image-guided surgery was considered unsuccessful if a single adenoma was found on the opposite side of the 4D-CT imaging result, if no adenoma was found during surgery, the intraoperative PTH level did not result in reduction of 50% after removal of this gland, or if there was persistent hypercalcemia or hyperparathyroidism postoperatively. 

The mean and standard deviation were used for all continuous variables and count and percentage for categorical variables. Statistical analyses were performed using SPSS software version 27.0 (IBM, Armonk, NY, USA). Values were considered statistically significant if the two-sided *p*-value was <0.05.

## 3. Results

### Patient Characteristics

A total of 60 patients with pHPT were included, consisting of 46 women and 14 men, with a mean age of 61.9 years (SD 10.9). Five patients had a history of neck surgery, of which three patients had undergone previous parathyroid surgery and had recurrent/persistent pHPT, and two patients had undergone previous thyroid surgery. Parathyroidectomy was performed in all 60 patients, including 46 patients with a single adenoma, 7 patients with a double adenoma, 1 patient with three adenomas, and 6 patients where no lesion was found during surgery. Baseline characteristics are provided in Table 1.

Twenty-eight patients had low baseline PTH levels (46,7%), and 32 patients had high baseline PTH levels (53.3%). Mean baseline PTH was 73.8 pg/mL in the low baseline PTH group compared to 247.0 pg/mL in the high baseline PTH group (*p* < 0.001). The mean preoperative serum calcium level was 10.7 mg/dL (2.67 mmol/L) in the low-PTH group compared to 11.2 mg/dL (2.79 mmol/L) in the high-PTH group (*p* = 0.03). The mean gland weight was 0.83 g in the low-PTH group compared with 1.4 g in the high-PTH group (*p* = 0.25). The incidence of MGD in the low baseline PTH group was 7.1% (2/28) compared to 18.8% (6/32) in the high baseline PTH group (*p* = 0.2). In the low baseline PTH group, two patients had a double adenoma. In the high-PTH group, five patients had a double adenoma, and one patient had three adenomas. 

Of 60 patients, 54 (90%) were surgically cured. In all patients in which no surgical cure was achieved, no lesion could be found during surgery and all patients had persistent postoperative hypercalcemia. A surgical cure was achieved in 23/28 patients (82.1%) in the low-PTH group. In the high-PTH group, 31/32 (96.9%) achieved a surgical cure. In the low baseline PTH group 5/28 (17.9%) patients did not achieve a surgical cure, compared to 1/32 (3.1%) patients in the high baseline PTH group (*p* = 0.06). A bilateral neck exploration was performed in one patient in the low-PTH group. In this patient, 4D-CT did not identify a candidate lesion pre-operatively. A bilateral neck exploration was performed in five patients in the high-PTH group, and all five patients achieved a surgical cure. 

## 4. Diagnostic Accuracy

In the low-PTH group, 4D-CT correctly localized 22 out of 25 lesions with a sensitivity of 88.0%, compared to 36 out of 38 lesions in the high-PTH group with a sensitivity of 94.7%. This difference was not statistically significant (*p* = 0.33). In the low-PTH group, 4D-CT identified a candidate lesion in 24 out of 28 patients (85.7%) in the low-PTH group and 31 out of 32 (97%) in the high-PTH group.

On a patient-based level, 4D-CT guided successful surgical resection in 20 out of 28 (71.4%) patients in the low baseline PTH group, compared to 29 out of 32 (90.6%) in the high baseline PTH group (*p* = 0.06). In the low baseline PTH group, surgical resection based on 4D-CT was unsuccessful in eight cases. In three of these cases, a parathyroid adenoma was found in a location not indicated by 4D-CT. However, all three patients achieved a surgical cure as the surgeon eventually resected the parathyroid adenoma. In five of the eight cases, no lesion was found during surgery. One of the five patients had recurrent pHPT. Four-dimensional dynamic CT guided successful surgical resection in both patients with MGD in the low-PTH group. Four-dimensional dynamic CT guided successful surgical resection in five out of six patients with MGD. In only one patient in the high-PTH group, a lesion could not be found during surgery. In the low-PTH group, there were two false-positive lesions, and both were in the same patient. Only one false-positive result was seen in the high-PTH group. The number of correct lesions identified in 4D-CT, and the lesion-based sensitivity is provided in Table 2. The image-guided surgical resection rate of 4D-CT in both patient groups is provided in Table 3. Patient and lesion characteristics of cases in which 4D-CT would not have led to successful image-guided surgery is provided in Table 4. 

Parathyroid adenoma visualization using four-phase 4D-CT. Adenoma clearly visible on left dorsal side of thyroid lobe. Adenoma exhibits typical enhancement characteristics through all phases. 

A: non-contrast phase (0 s);

B: arterial phase (30 s);

C: venous phase (50 s);

D: delayed phase (90 s).

## 5. Discussion

The objective of this study was to assess the diagnostic accuracy of 4D-CT in patients with low baseline PTH compared to patients with high baseline PTH. A total of 60 patients with pHPT were included in this retrospective study, comprising 28 patients in the low baseline PTH group and 32 patients in the high baseline PTH group. The results of this study demonstrated a high lesion-based sensitivity of 4D-CT in both study groups, with a sensitivity of 88% for patients with low baseline PTH levels, compared to 94.7% for patients with high baseline PTH levels (*p* = 0.33). However, 4D-CT led to successful image-guided resection in only 71.4% of patients in the low baseline PTH group, compared to 90.6% in the high baseline PTH group (*p* = 0.06). 

In recent years, the rise in neck imaging studies and calcium measurements for osteoporosis screening has contributed to a growing trend of diagnosing patients with pHPT, including patients with low baseline levels of PTH [4]. As low baseline levels of PTH are associated with a higher rate of MGD, unsuccessful minimally invasive parathyroidectomies, and lower localization rates of preoperative imaging modalities, accurate preoperative localization of parathyroid lesions is crucial in order to achieve successful minimally invasive surgery [2,3,5,6,7,8]. Hence, in patients with low baseline PTH levels, precise preoperative localization of parathyroid lesions becomes even more critical, ensuring the successful surgical resection of all identified parathyroid lesions. 

The lesion-based sensitivity in this study is in line with a previous study that reported a high sensitivity of 84.6% for 4D-CT in patients with low baseline PTH levels [5]. However, the previous study did not compare the diagnostic accuracy in patients with high baseline PTH [5]. In this study, 4D-CT only guided successful surgical resection in 71.4% of patients with low baseline PTH levels, compared to 90.6% in patients with high baseline PTH levels. This result is in concordance with several studies that have shown lower diagnostic accuracy of preoperative imaging modalities such as MIBI and ultrasound in patients with low baseline PTH levels [2,7]. For example, one study found that the localization rate of MIBI was 68.9% in patients with low baseline PTH, compared to 83.7% in patients with high baseline PTH [2]. 

Among patients with low baseline PTH levels, surgical resection based on 4D-CT localization would have been unsuccessful in eight cases (29%). In five of these cases, (18%), no parathyroid lesion was found during surgery, and all patients had persistent postoperative elevated levels of PTH and/or calcium. In comparison, in only one patient (3.1%) in the high baseline PTH group, no lesion was found during surgery. This result is in line with a study that showed that lower preoperative PTH levels are an independent risk factor for operative failure in parathyroidectomy [8]. The possible reasons for the higher rate of operative failure in patients with low baseline PTH levels are multifactorial. First, these patients may have a high incidence of MGD [3,5,6,24,25,26], and parathyroidectomy failure occurs at higher rates in patients with MGD [27]. Second, several studies have shown that the preoperative localization rates of imaging modalities are lower in these patients, as the lesions tend to be smaller in size and weight [2]. Third, IOPTH assays may have a lower specificity and a higher false-negative rate in these patients, as the baseline PTH levels are already low [28]. It should be noted that the success rate of surgical resection depends not only on the accuracy of the preoperative imaging modality but also on the surgeon’s skill. All these factors may contribute to a higher chance of missing parathyroid lesions during surgery.

Several studies have found that the rate of MGD in patients with low baseline PTH levels is higher than patients with high baseline PTH [3,6,29,30]. Interestingly, in this study, only 7.1% of patients with a low baseline PTH had MGD compared to 18.8% in the high baseline PTH group. However, this rate did not differ significantly between the groups (*p* = 0.19). This finding is most likely a random finding that can possibly be attributed to the small number of patients in these groups. Furthermore, it is a possibility that MGD was present in the patients in which no lesion was found. 

Multiple studies have found that 4D-CT has a high diagnostic accuracy including in patients with recurrent pHPT and patients with MGD [5,13,14,16]. However, a significant drawback of 4D-CT is the relatively high radiation exposure that results from the multiple scanning phases [23,31,32,33,34,35]. This has led to concerns about the potential for an increased risk of thyroid cancer, particularly in younger patients [31]. Therefore, it has been argued by some authors to use 4D-CT judiciously, especially in younger patients [31]. Currently, there is no consensus on the optimal number of phases for the best balance between radiation exposure and a high diagnostic yield [36]. However, the implementation of virtual non-contrast images using dual-energy CT might reduce radiation exposure by eliminating the need for a true non-contrast phase [37,38]. 

In addition to 4D-CT, various emerging diagnostic modalities are currently under investigation for the localization of PHPT. These novel techniques include radiolabeled positron emission tomography CT, dynamic contrast-enhanced magnetic resonance imaging, and elastography [39,40,41]. Studies have demonstrated that these modalities hold great promise in improving the accuracy of PHPT localization and may offer certain advantages over 4D-CT, particularly concerning radiation exposure to patients [41,42,43]. While these diagnostic modalities have demonstrated high diagnostic accuracy in localizing PHPT, it is essential to recognize that their value in patients with low baseline PTH levels remains largely unexplored. Since the detection of parathyroid lesions may be more challenging in patients with low PTH, investigating the sensitivity and specificity of these emerging techniques in such cases is crucial. Future research efforts should focus on evaluating the performance of these emerging diagnostic modalities in patients with low baseline PTH levels. Additionally, direct head-to-head comparative studies with 4D-CT in different patient populations will elucidate the relative benefits and limitations of each diagnostic modality.

## 6. Limitations

This study may be subject to limitations that need to be considered. First, it is a retrospective single-center study with a relatively small sample size, which may limit the generalizability of our results. However, this study represents the largest cohort to date that evaluates 4D-CT in low baseline PTH patients. Second, localizing parathyroid adenomas is dependent on the surgeon’s skill and experience. Therefore, the rate of image-guided resection based on 4D-CT results may differ between surgeons and centers. Moreover, the distinction between a mediastinal and a lower cervical gland may be subjective and dependent on the information available in the surgical reports. 

## 7. Conclusions

In conclusion, this study demonstrated that 4D-CT has a high lesion-based sensitivity in patients with primary hyperparathyroidism and low baseline PTH levels but led to a relatively low rate of successful image-guided resection in patients with low baseline PTH levels. Therefore, it is important to exercise increased caution during 4D-CTguided surgical exploration of patients with low baseline PTH levels to ensure successful surgical resection of all parathyroid lesions. Further prospective studies are needed to validate the results of this study and to investigate the value of 4D-CT in patients with low baseline PTH. 

## Figures and Tables

**Figure 1 diagnostics-13-02621-f001:**
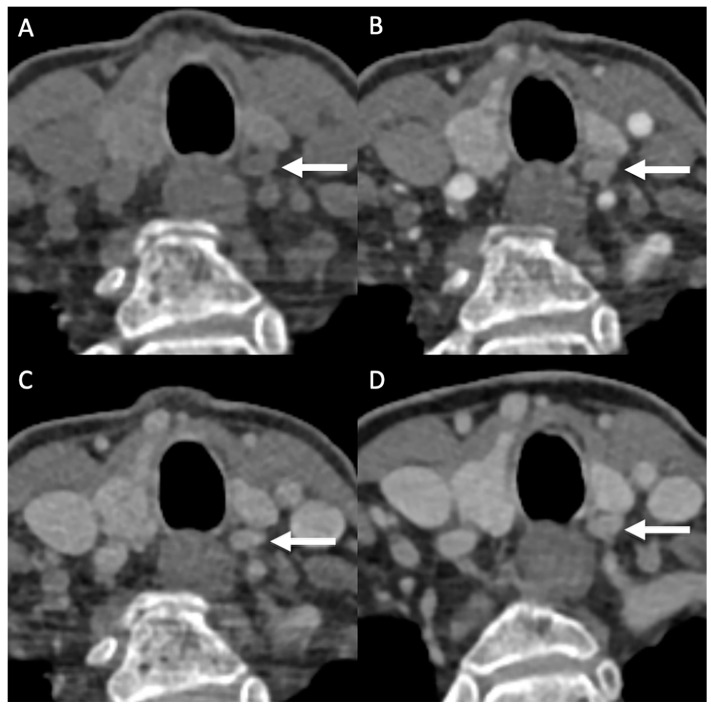
Visualization of parathyroid adenoma in 4D-CT using 4 phases. (**A**) Non Contrast CT scan. (**B**) Contrast Enhanced CT scan after 30 s of Contrast administration. (Arterial Phase) (**C**) Contrast Enhanced CT scan after 50 s of Contrast administration. (Venous Phase) (**D**) Contrast Enhanced CT scan after 90 s of Contrast administration. (Delayed Phase).

**Table 1 diagnostics-13-02621-t001:** Summary of patient characteristics per study group.

Characteristic	Low PTH (n = 28)	High PTH (n = 32)	*p*-Value
Female	23 (82.1%)	23 (71.9%)	0.42
Age	61.1 (9.3)	61.7 (12.3)	0.74
BMI	26.5 (4.4)	26.6 (5.7)	0.6
Baseline PTH	73.8 pg/mL (SD 17.9)	247.0 pg/mL (241.7)	<0.001
Baseline Calcium	10.7 mg/dL (0.5)	11.2 mg/dL (0.8)	0.03
SGD	92.9% (26/28)	87.5% (28/32)	-
MGD	7.1% (2/28)	18.8% (6/32)	0.19
Double adenoma	2	5	
Triple	0	1	-
Weight	0.77 g (SD 0.69)	1.4 g (SD 1.6)	0.25
Max diameter	15.9 mm (8.6)	19.5 mm (10.9)	0.23
Lesion not found during surgery	5 (17.9%)	1 (3.1%)	0.058
BNE	1	5	0.12

BMI: body mass index, BNE: bilateral neck exploration, MGD: multi-gland disease, PTH: parathyroid hormone, SD: standard deviation.

**Table 2 diagnostics-13-02621-t002:** Lesion-based sensitivity rate per study group.

Parameter	Low PTH	High PTH	*p*-Value
Correct localization (N)	22	36	-
Total glands (N)	25	38	-
Lesion-based sensitivity	88.0% (22/25)	94.7% (36/38)	0.33

PTH: parathyroid hormone.

**Table 3 diagnostics-13-02621-t003:** Successful image-guided surgical resection rate per study group.

Parameter	Low PTH	High PTH	*p*-Value
Correct localization (N)	20	29	-
Total glands (N)	28	32	-
Successful image-guided surgical resection rate	71.4% (20/28)	90.6% (29/32)	0.056

PTH: parathyroid hormone.

**Table 4 diagnostics-13-02621-t004:** Patient and lesion characteristics of cases in which 4D-CT findings did not guide successful surgical cure.

No.	4D-CT Finding	Previous Neck Surgery	Thyroid Comorbidity	Preoperative PTH (pg/mL)	Preoperative Calcium (mg/dL)	Adenoma Weight (grams)	Surgery/Histopathology	Cure
1	1 lesion identified: right side	no	no	99.02	10.40	-	No lesion found during surgery	no
2	1 lesion identified: intrathyroidal right side	Yes: previous parathyroidectomy	no	41.49	10.36	-	No lesion found during surgery	no
3	1 lesion identified: left side	no	no	68.84	10.88	-	No lesion found during surgery	no
4	1 lesion identified: right side	no	no	84.87	10.96	-	No lesion found during surgery	no
5	No lesion identified	no	no	99.02	10.36	0.40	1 parathyroid adenoma left side	yes
6	No lesion identified	no	no	96.19	10.32	0.32	1 parathyroid adenoma left side	yes
7	No lesion identified	no	no	36.78	10.24	0.40	1 parathyroid adenoma right side	yes
8	1 lesion identified: left side	no	no	139.56	12.72	6.5 and 0.48	2 parathyroid adenomas: left side and right side.	yes
9	No lesion identified	no	no	68.84	11.00		No lesion found during surgery	no
10	1 lesion identified: mediastinal	no	no	207.46	22.00		No lesion found during surgery	no
11	No lesion identified	no	no	809.09	12.52	3	1 parathyroid adenoma found in thymus	yes

## Data Availability

The datasets generated or analyzed during the current study are included in this published article.

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
