# Peer review of "Diagnostic Value of Four-Dimensional Dynamic Computed Tomography for Primary Hyperparathyroidism in Patients with Low Baseline Parathyroid Hormone Levels"

_diagnostics, 2023, doi:10.3390/diagnostics13162621_

Round 1
Reviewer 1 Report
A very well-done article, factors that prevent its submission for publication are not identified. My congratulations to the authors. I have only two remarks to make:
Line 18: The acronym PTH appears without its meaning appearing, the same happening with the other acronyms that appear for the first time in the text,
Line 87: The authors must indicate in which situations, during the first phase of the study, DECT was performed and in which situations SECT non-contrast scan was performed, and the reason for this variation in the protocol.
Once again, congratulations to the authors
Author Response
Thank you for your feedback on our manuscript.
We would like to thank the reviewers for reviewing the article and providing helpful feedback that has strengthened the manuscript.
We have added a paragraph to the discussion section and the total word count has now reached 4040 words starting from the abstract to the conflict-of-interest statement. We hope the following revisions are satisfactory and we look forward to hearing if there are any additional suggestions to strengthen the manuscript.
Reviewer 1
Comment: A very well-done article, factors that prevent its submission for publication are not identified. My congratulations to the authors. I have only two remarks to make:
Answer: We thank you for your excellent feedback on the article
Comment: Line 18: The acronym PTH appears without its meaning appearing, the same happening with the other acronyms that appear for the first time in the text,
Answer: Thank you for pointing this out. Per your feedback, we have now written every acronym fully before abbreviating the term throughout the manuscript.
Comment: Line 87: The authors must indicate in which situations, during the first phase of the study, DECT was performed and in which situations SECT non-contrast scan was performed, and the reason for this variation in the protocol.
Answer: Thank you for this very valid feedback. The initial scanning protocol consisted of a SECT non-contrast scan because initially, we deemed it unnecessary to perform a DECT non-contrast scan as we initially thought that this would not be of added benefit. However, we changed our scanning protocol to perform DECT non-contrast scanning because we also wanted to determine the iodine content of the thyroid on the non-contrast scans using DECT for other research purposes. In this research protocol, 31 patients had a SECT non-contrast scan and 29 patients had a DECT non-contrast scan.
Comment: Once again, congratulations to the authors
Answer: We thank you for your feedback on the article.
Reviewer 2 Report
The stdy is very interesting with important conclusions.
The objective of the study is to assess the diagnostic value of 4D-Ct in localizing pHPT in patients with low baseline PTH levels, comparet with patients with high baseline PTH levels.
The study demonstrated that 4D-CT has a high lesion-based sensitivity in patients with pHPT and low baseline PTH levels. There is important to exercise increased causion during 4D-CT guided surgical exploration of patients with low baseline PTH levels to ensure successful surgical resection of all patathyroid lesions.
Author Response
Thank you for your feedback on our manuscript.
We would like to thank the reviewers for reviewing the article and providing helpful feedback that has strengthened the manuscript.
We have added a paragraph to the discussion section and the total word count has now reached 4040 words starting from the abstract to the conflict-of-interest statement. We hope the following revisions are satisfactory and we look forward to hearing if there are any additional suggestions to strengthen the manuscript.
Reviewer 2
Comment: The stdy is very interesting with important conclusions.
The objective of the study is to assess the diagnostic value of 4D-Ct in localizing pHPT in patients with low baseline PTH levels, comparet with patients with high baseline PTH levels.
The study demonstrated that 4D-CT has a high lesion-based sensitivity in patients with pHPT and low baseline PTH levels. There is important to exercise increased causion during 4D-CT guided surgical exploration of patients with low baseline PTH levels to ensure successful surgical resection of all patathyroid lesions.
Answer: We thank you for your feedback!
Reviewer 3 Report
In the manuscript " Diagnostic Value of 4D-CT for Primary Hyperparathyroidism in Patients with Low Baseline Parathyroid Hormone Levels", the authors present a very interesting study on the utility of 4D CT for the identification of parathyroid adenomas in cases of primary hyperparathyroidism. The paper is well written, and the facts are well presented, however, minor criticisms are present, as follows:
- There title should be changed to Diagnostic Value of 4D-CT for Primary Hyperparathyroidism, as the utility in the case of low PTH can be datable, as shown in the table 4.
- There are discussions about the role of elastography in the evaluation of hyperparathyroidism and how it can be used as a complementary technique in the case of identifying difficult parathyroid adenomas.
- The limitations are well presented.
- The conclusions should be adapted to the title, but otherwise well presented.

Author Response
Thank you for your feedback on our manuscript.
We would like to thank the reviewers for reviewing the article and providing helpful feedback that has strengthened the manuscript.
We have added a paragraph to the discussion section and the total word count has now reached 4040 words starting from the abstract to the conflict-of-interest statement. We hope the following revisions are satisfactory and we look forward to hearing if there are any additional suggestions to strengthen the manuscript.
Reviewer 3
In the manuscript " Diagnostic Value of 4D-CT for Primary Hyperparathyroidism in Patients with Low Baseline Parathyroid Hormone Levels", the authors present a very interesting study on the utility of 4D CT for the identification of parathyroid adenomas in cases of primary hyperparathyroidism. The paper is well written, and the facts are well presented, however, minor criticisms are present, as follows:
Comment: Their title should be changed to Diagnostic Value of 4D-CT for Primary Hyperparathyroidism, as the utility in the case of low PTH can be datable, as shown in the table 4.
Answer: per your suggestion, we have changed the title of the manuscript.
Comment: There are discussions about the role of elastography in the evaluation of hyperparathyroidism and how it can be used as a complementary technique in the case of identifying difficult parathyroid adenomas.
Answer: Thank you for this valuable suggestion. We have added a paragraph in the discussion section outlining other promising diagnostic modalities including elastography. We have added the following section from line 274 to line 288:
“In addition to 4D-CT, various emerging diagnostic modalities are currently under investigation for the localization of PHPT. These novel techniques include radiolabeled positron emission tomography-CT, dynamic contrast-enhanced magnetic resonance imaging, and elastography. Studies have demonstrated that these modalities hold great promise in improving the accuracy of PHPT localization and may offer certain advantages over 4D-CT, particularly concerning radiation exposure to patients. While these diagnostic modalities have demonstrated high diagnostic accuracy in localising PHPT, it is essential to recognize that their value in patients with low baseline PTH levels remains largely unexplored. Since the detection of parathyroid lesions may be more challenging in patients with low PTH, investigating the sensitivity and specificity of these emerging techniques in such cases is crucial. Future research efforts should focus on evaluating the performance of these emerging diagnostic modalities in patients with low baseline PTH levels. Additionally, direct head-to-head comparative studies with 4D-CT in different patient populations will elucidate the relative benefits and limitations of each diagnostic modality.”
Comment: The limitations are well presented.
Answer: We thank you for your feedback
Comment: The conclusions should be adapted to the title, but otherwise well presented.
Answer: Thank you for the feedback. We have now expanded the last sentence of the conclusion to fit the title:
“Further prospective studies are needed to validate the results of this study and to investigate the value of 4D-CT in patients with low baseline PTH.”
